# Perceptions of Cultural Ecosystem Services in Urban Parks Based on Social Network Data

**Peichao Dai** **, Shaoliang Zhang \*** , **Zanxu Chen, Yunlong Gong and Huping Hou**

School of Environment Science and Spatial Informatics, China University of Mining and Technology, 1st Daxue Road, Xuzhou 221000, China; dpc@cumt.edu.cn (P.D.); chenzanxu@cumt.edu.cn (Z.C.); ylgong@cumt.edu.cn (Y.G.); hphou@163.com (H.H.)
* Correspondence: slzhang@cumt.edu.cn

**Abstract:** The value of a cultural ecosystem service depends on the perception of different cultural service categories. However, the data sources used in research on the perception of cultural service have limitations that mainly depend on social investigation, leading to slow progress in cultural service evaluation. With the advent of the era of network big data, social media provides a new data source for the study of cultural ecosystem service perception, so that the study of these services is expected to make new breakthroughs. Using search crawler software, this paper reviewed 7257 online comments related to 19 city parks in Xuzhou City, China. With the help of Rost Content mining semantic analysis software, the comment sentences were divided into keywords, and the Delphi expert method was used to classify these keywords. Thus, a cultural service perception database was established. Through statistical analysis, with the help of ArcGIS software, various cultural services were analyzed. The results showed that (1) the cultural services of urban parks could be divided into seven types (i.e., aesthetics, recreation, sports, inspiration, education, cultural heritage, and spiritual satisfaction) using social network comment data. (2) High-frequency keywords of online comment data can serve as the core basis during an analysis of the perception of cultural services by visitors of city parks. However, a large gap exists in the number of high frequency keywords in different parks. For example, Yunlong Lake Park has 2887 keywords, while Kuaizai Ting Park has only 33. (3) Differences exist in the perception of cultural service in urban parks, the park's scale, and characteristics determine the visitor's cultural service perception level. The aesthetic and recreation types were the most easily perceived, and 68% and 63% parks have the above two perceptual records, respectively. Therefore, the social media comment data has the ability to document perception of each park's cultural service type and its differences, which can serve as the cultural ecosystem service perception as well as the valuation data source, to supplement the social investigation.

**Keywords:** cultural ecosystem services; urban parks; perception; social network data

---

## 1. Introduction

With 55% of the world's population living in cities in 2018, a number that is expected to grow to 68% by 2050 [1], the ability of urban ecosystems to provide adequate ecosystem services to so many residents is an issue that needs to be addressed in urban ecology research. Urban ecosystem services are often related to the perception of urban residents because these residents believe that urban parks serve as important places for urban residents to enjoy recreation [2] and environmental aesthetics [3]. In addition, these parks serve as the main providers of cultural ecosystem services and related products, such as aesthetics, recreation, tourism, education, and spiritual experiences [4,5]. Nevertheless, different groups may perceive these cultural services differently. Therefore, research into

the perception of cultural services by urban residents has significant meaning for urban park planning, biodiversity conservation, and human health [6,7].

The cultural ecosystem service depends on the subjectivity of human perception to realize its ultimate value [8]. However, the variation in this perception among different types of people is difficult to quantify. Older adults, for example, prefer cultural ecosystem services related to natural experiences, while urban youth tend to socialize [9]; park staff perceive park services more professionally, while laymen seem to focus more on superficial landscapes; and experts' perceptions of nature appear to be more management-centered, whereas laypersons appear to prioritize enjoyment of nature [10]. Thus, different groups have different perceptions of cultural services in urban parks.

At present, the different perceptions of cultural ecosystem services are mostly studied using questionnaires [11,12], intercept surveys [13], photographic analysis [14,15], software (e.g., SolVES) [16], or visitor-employed photography with a short comments [17]. With the rise of social media, social networks have gradually become an important method of obtaining assessment data [18]. That is, social networking sites designed for tourists provide a new way to study perceptions of cultural ecosystem services. Online review data from social networks are more numerous and easier to collect than the data acquired using questionnaires.

The data from online social networks on travel websites include the impressions of tourists who enjoyed their leisure and recreation activities in an urban park. Tourist comments may include feelings, experiences, and photographs [19]. These data and comments are influenced by the subjects' gender, age, occupation, educational background, economic conditions, and other factors. They can reflect the commentator's perceived state and emotional attitude towards a specific park. Currently, research using online shopping review data for consumer behavior analysis has contributed some preliminary achievements [20,21], nevertheless, the use of tourism website online comment data for cultural ecosystem services perception research has rarely been reported.

In this paper, we demonstrate that data from social media networks offer new opportunities to investigate perceptions of cultural ecosystem services in urban parks. Using 19 urban parks in Xuzhou as an example, this paper analyzes the online comments of social networks as a data source. This study addressed three questions. First, how does online comment data match the various categories of cultural ecosystem services? Second, can the types of cultural ecosystem services provided by each park be determined based on online comments? Third, how can differences in the perception of the cultural ecosystem services provided by each park be quantified based on comment data?

## 2. Methods and Materials

### Study Area

Xuzhou is located in northwestern Jiangsu Province in the developed region of Eastern China (33°43′–34°58′ N, 116°22′–118°40′ E, Figure 1). The history of civilization in the Xuzhou area extends back more than 6000 years, while its history as a city goes back more than 2600 years. Xuzhou is known as the cultural birthplace of both the Eastern and Western Han dynasties. Xuzhou was honored as a National Forest City in 2012, National Ecological Garden City in 2015, and the Habitat Scroll of Honor Award in 2018. By the end of 2017, Xuzhou City had developed 83 urban parks with a built-up area greening coverage of 43.84%, green space rate of 41.06%, and per capita park green area of 15.08 m$^2$. Several parks, including Chuwang Tomb in Han Culture Park, the Han Tomb in Guishan Park, the Aquarium Exhibition Hall in Yunlong Lake Park, and the zoo in Pengzu Park charge admission fees, however, all of the other parks are completely open to everyone.

Considering the amount of online comment data available, this is work is based on the 19 largest of the 83 parks, each with over 30 comment records. The park locations are shown in Figure 1.

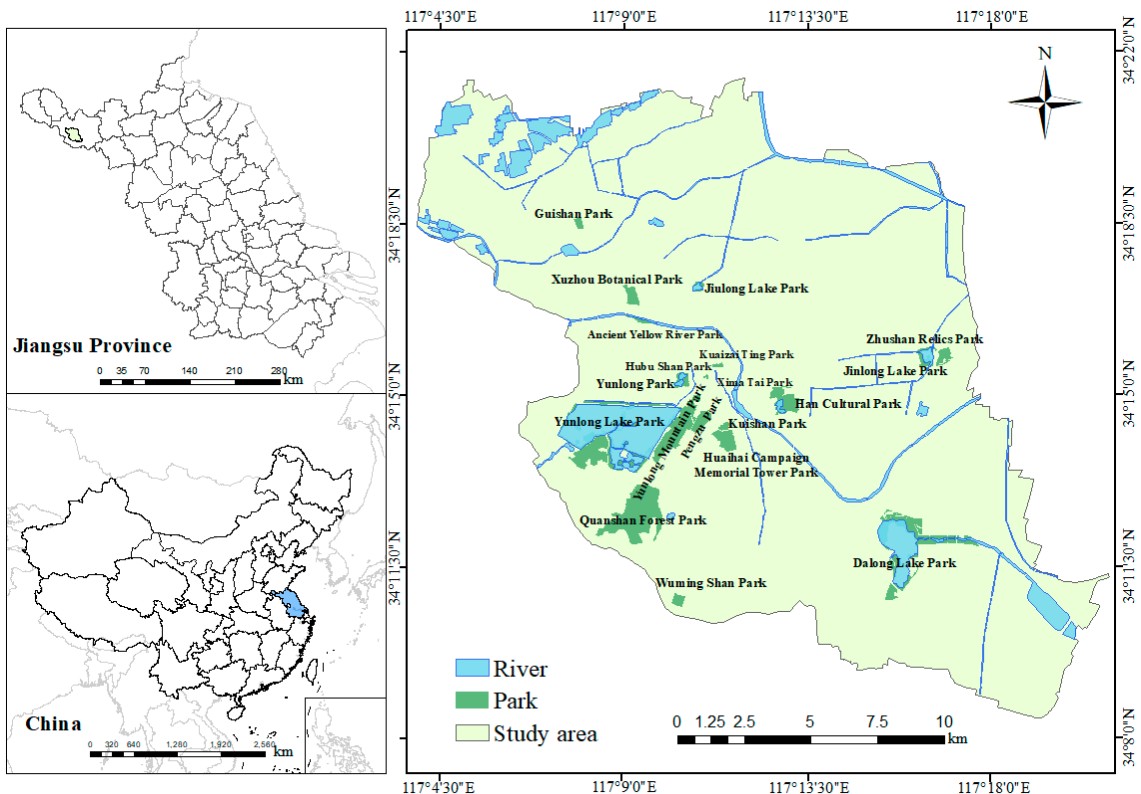

**Figure 1.** Map of Xuzhou and the distribution of 19 urban parks.

## 3. Data Sources and Methods

### 3.1. Data Sources

The data of this paper are derived from three online travel comment portals frequently used in China, namely Ctrip Travel [22], Baidu Travel [23], and Tuniu Travel [24] websites. These websites are the most popular ones in China, especially Ctrip Travel, which is China's largest online travel website with 135 million users and the greatest number of comment records in 2018. The design of the comment area interface is different in each of these three websites. However, all major functions have similarities such as using username, comment publication date, comment content, scoring evaluation (Ctrip Travel: Very good, good, general, poor, very poor. Baidu Travel and Tuniu Travel networks: 5 stars, 4 stars, 3 stars, 2 stars, 1 star), and picture sharing functions.

Using Gooseeker search crawler software, the comment data (in Chinese) provided by users of 19 parks in Xuzhou from 17 April 2012 to 28 September 2018 were collected from the above three tourist social networking sites, including 7257 comments (Figure 2) and 323,189 words. Then, all these data were imported into a database, the comment content was sorted alphabetically, and 135 duplicate reviews and invalid comments were deleted (Figure 3).

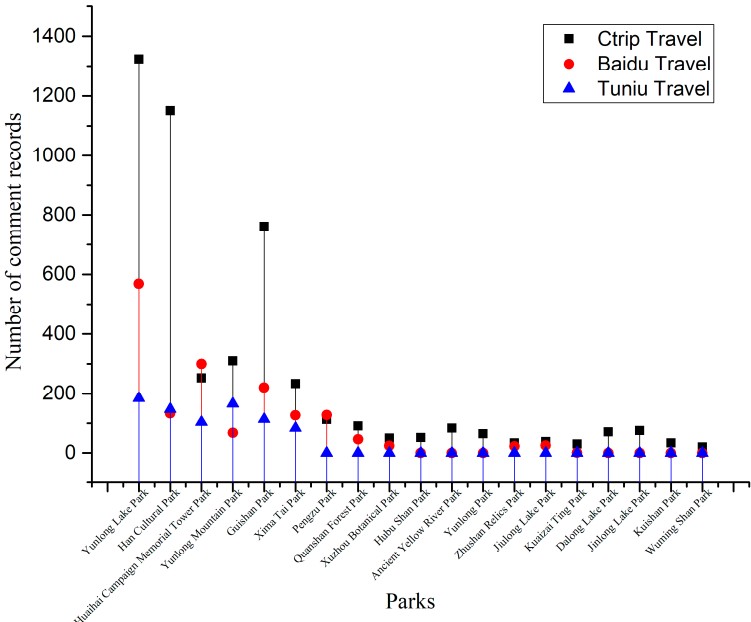

**Figure 2.** Records of comments related to parks on three websites (Ctrip, Baidu, and Tuniu).

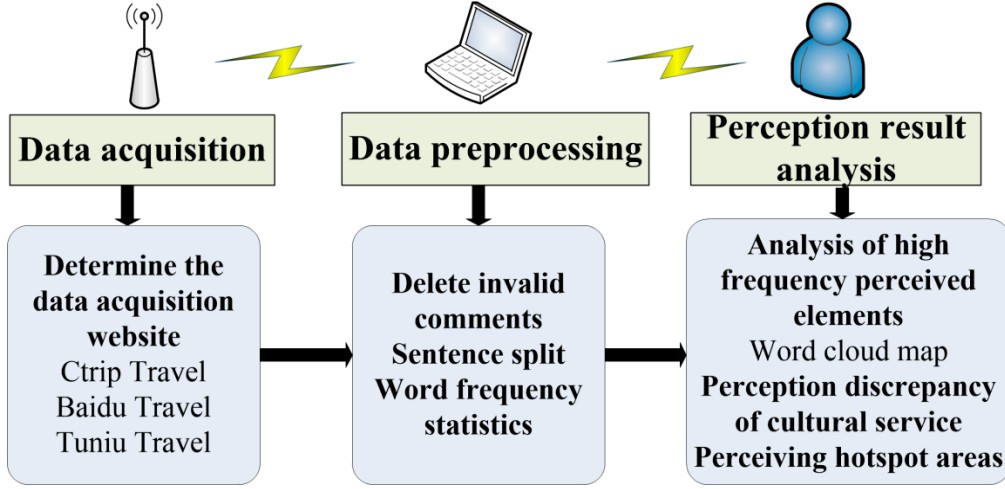

**Figure 3.** Research framework of cultural service perception.

*3.2. Analysis of Online Comment Data*

First, each sentence was divided into independent words by using ROST Content Mining software. Next, the occurrence frequency of each word was analyzed, providing the top 200 most frequently used words for each park. Words that were not related to the subject or that had no clearly point, such as "fit", "located", "around", "have been", and so on were filtered out and not analyzed further. Then, the perception corpus of cultural service was established.

Second, the Delphi method was used to classify the words making up the comments. The Chinese language uses a complex system of forming words that often involve two or more Chinese characters making the language itself complex. Therefore, even though common morphemes exist between words, the meaning of each expression is often very different. For example, "美妙" means wonderful, while "美国" means the United States in English. However, while the common morpheme "美" means beauty or beautiful in English, the two phrases have distinctly different meanings. This paper employed ten experts and used the Delphi expert evaluation method to combine the characteristics of various cultural service categories. The keywords extracted from the comment data were perceived and correlated with the category of cultural services. This method comprised the following steps (Figure 4):

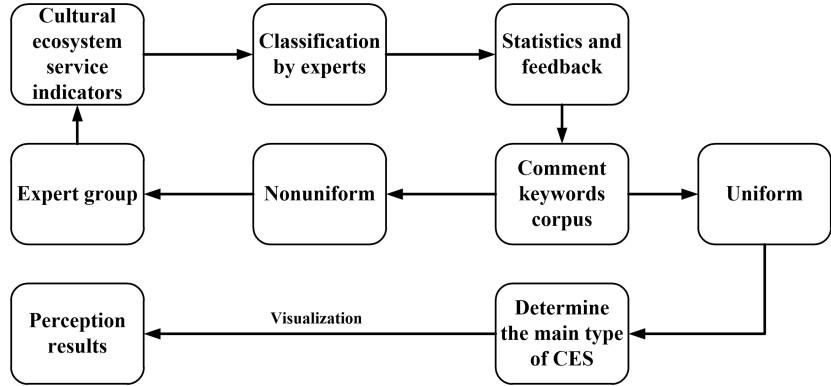

**Figure 4.** The steps of research design.

(1) Built a team of professional subject experts in the field of ecology and land resource management.

(2) Established classification indicators for each of the cultural ecosystem services provided by the urban parks. We referred to the relevant literature related to cultural ecosystem service classification for urban parks [25], combined the high frequency perceived elements extracted from park visitor comments, and constructed a classification system for the cultural services provided by the urban parks. According to the classification based on comment keywords, the cultural services of urban parks were divided into seven categories (aesthetics, recreation, sports, inspiration, education, cultural heritage, and spiritual satisfaction).

(3) Classification by experts. Next, the established comment keywords and classification table were sent to experts in the form of e-mail. The experts could then begin their first round of classification into to the seven types based on their professional knowledge and return the classification results.

(4) Statistics and feedback. After receiving the results of the first review of the classification from all experts, the results were summarized. The comment keywords of the same classification were extracted, and the words classification that was not relatively uniform was marked. Next, the results were resubmitted to the experts.

(5) The second-round classification. After the results of keyword categories that were not uniform in the first-round classification were resubmitted to the experts, a second round of classification adjustment was carried out and new classification results were created.

(6) Make the classification uniform. After receiving the results of the second-round classification from the experts, steps (3) and (4) were repeated to unify the keyword classification until a final unified opinion was formed. This research for the present article involved three rounds of feedback that formed the final classification result for keywords, that is, the comment keyword corpus.

(7) The comment keywords that match each park were summarized with the comment keywords corpus based on the frequency of each keyword used by park visitors.

(8) Based on the number of comment keywords left by visitors for each cultural service type of each park, the perceived results were summarized for each park individually to determine the main types of cultural service provided, as well as to identify the differences in the cultural services within each park.

(9) Based on the classification of cultural services in each park, the perception results of each park were visualized by using ArcGIS (Esri: Redlands, CA, USA).

## 4. Results

### 4.1. Perception Correlation between Keywords and Cultural Services

Many studies that classified cultural services have been previously devised. Researchers have employed various names for cultural ecosystem services. Some were called "cultural service" [26], "information function" [27], "delightful and satisfaction" [28], or "social and cultural achievement" [29].

There are subdivided into their intrinsic functions from the point of view of satisfying the needs of the human spirit. In the present study, the cultural services were classified into seven types according to the characteristics of the research area or the research object. Therefore, based on referring to the relevant published literature [30], combined with the actual situation of parks in the study area and the content of the comment data, the cultural service classification of the urban parks was constructed (Table 1).

**Table 1.** Cultural ecosystem service categories in urban parks.

| Cultural Service Categories | Description of Cultural Service | Representative Keywords |
|---|---|---|
| Aesthetic | Beautiful natural landscape of park | Beautiful, beauty, scenery, good-looking |
| Recreation | Park offers tours and relaxing places | Explore, fun, walk, boating, entertainment, sightseeing |
| Sports | Park provides sports venues and facilities | Exercise, running, fitness, morning exercise, swimming, mountain climbing |
| Inspiration | Park provides material for artistic creation | Legends, imaginations, stories, music, art, culture |
| Education | Park broadens people's knowledge | Visits, education, exhibitions, admiration, remembrance |
| Cultural heritage | Preservation of historical and cultural heritage | Monuments, cultural relics, tombs, antiques, passageway, palaces |
| Spiritual satisfaction | People's pleasurable mood in the park | Striking, spectacular, relaxed, romantic, exquisite, magical |

Keywords were identified as follows. (1) The keywords in the aesthetic category were primarily adjectives, describing the beautiful scenery of a park, including beauty, good-looking, and nice. (2) The keywords in the recreation category were mostly related to recreational behavior, including walking and leisure. (3) The keywords in the sports category were mainly verbs, related to sport activity, including running and exercise. (4) The keywords in the inspiration category were mainly related to artwork and inspirational thoughts, including legend and music. (5) The keywords in the educational category were related to the method of educating park visitors, such as a visitor obtaining knowledge through a narrator's explanation. (6) The keywords in the cultural heritage category were closely related to the regional characteristics, as Xuzhou has numerous tombs from the Han Dynasty; thus, Han tombs, cultural relics, terracotta warriors, and other keywords were categorized as cultural heritage. (7) The keywords in the spiritual satisfaction category mainly reflected the commentator's psychological state, including romantic and relaxed.

*4.2. Analysis of Perception Results of Cultural Ecosystem Services*

4.2.1. High-Frequency Keywords Reflect the Perception Results

Table 2 lists the top ten most frequently used words that visitors used to describe the 19 parks of Xuzhou, which were collated according to the online comments, and can be seen as a good reflection of the park's cultural service. For example, the beautiful scenery of Yunlong Lake Park was often compared with the West Lake Cultural Landscape of Hangzhou; therefore, the words "West Lake" and "beautiful" appear frequently. In addition, Han Cultural Park is an open park characterized by the mausoleum of King Chu in the western Han Dynasty, for which the phrases "Terracotta Warriors" and "Han Culture" represent this time period and were frequently employed words. Parks in more densely populated cities, such as the parks called Yunlong Lake, Jiulong Lake, Kuaizai Ting, Kuishan, and Wuming Shan, provide important outdoor spaces for residents to exercise; therefore, high frequency words used for these parks include, "exercise", "walking", "running", and other such keywords.

**Table 2.** The top ten high-frequency words of each park.

| Park | | | | | | | | | | | |
|---|---|---|---|---|---|---|---|---|---|---|---|
| **Yunlong Lake Park** | Characteristics words | Yunlong Lake | Scenery | Xuzhou | Scenic Spots | Place | West Lake | Yunlong Mountain | Park | Environment | Attractive |
| | Frequency | 707 | 610 | 552 | 393 | 256 | 231 | 195 | 183 | 161 | 158 |
| **Han Cultural Park** | Characteristics words | Terracotta warriors | Xuzhou | Scenic spot | The Chuwang tomb | Han culture | Scenic Spots | Museum | History | Culture | Shizi Shan |
| | Frequency | 699 | 428 | 380 | 371 | 284 | 244 | 205 | 196 | 182 | 175 |
| **The Huai Hai Campaign Memorial Tower Park** | Characteristics words | Huai Hai Campaign | Memorial | History | Memorial Tower | Xuzhou | Martyr | Education | Campaign | Place | Free |
| | Frequency | 330 | 147 | 137 | 126 | 123 | 118 | 114 | 93 | 89 | 85 |
| **Yunlong Mountain Park** | Characteristics words | Yunlong Mountain | Xuzhou | Yunlong Lake | Ropeway | Scenery | On the hill | Scenery | Han Portrait Stone | Peak | History |
| | Frequency | 318 | 210 | 110 | 85 | 83 | 65 | 63 | 61 | 61 | 47 |
| **Guishan Park** | Characteristics words | Guishan Tomb | Xuzhou | Imperial edict | Scenic Spots | Museum | Han dynasty tombs | History | Scenic spot | Worth watching | Ancients |
| | Frequency | 282 | 272 | 229 | 207 | 205 | 185 | 170 | 146 | 137 | 120 |
| **Xima tai Park** | Characteristics words | Xima Tai | Xiangyu | History | Xuzhou | Place | Scenic Spots | Overlord | Drama Horse | Western Chu | Hubu Shan |
| | Frequency | 152 | 149 | 112 | 104 | 82 | 77 | 72 | 68 | 51 | 50 |
| **Pengzu Park** | Characteristics words | Xuzhou | Pengzu | Zoo | Sakura | Pengzu Park | Park | Animal | Place | Free | Child |
| | Frequency | 101 | 83 | 55 | 52 | 51 | 46 | 34 | 30 | 27 | 23 |
| **Quanshan Forest Park** | Characteristics words | Quanshan Forest | Park | Xuzhou | Place | Suitable | Scenic spot | Scenery | Free | Summer | Environment |
| | Frequency | 47 | 42 | 27 | 23 | 16 | 14 | 14 | 13 | 12 | 12 |
| **Xuzhou Botanical Park** | Characteristics words | Plant | Botanical garden | Place | Tickets | Xuzhou | Tropical | Recreation | Flower | Species | Place |
| | Frequency | 44 | 26 | 17 | 11 | 10 | 10 | 9 | 9 | 8 | 6 |
| **Hubu Shan Park** | Characteristics words | Xuzhou | Hubu Shan | Xima Tai | Culture | Place | Courtyard | Architecture | History | Rich | Curio |
| | Frequency | 33 | 26 | 16 | 13 | 11 | 11 | 10 | 9 | 8 | 7 |
| **Ancient Yellow River Park** | Characteristics words | Xuzhou | Ancient Yellow River | Yellow River | Good | Park | Scenic spot | Scenery | Evening | Xuzhou City | By the River |
| | Frequency | 30 | 26 | 21 | 18 | 17 | 12 | 11 | 9 | 8 | 8 |
| **Yunlong Park** | Characteristics words | Park | Free | Yunlong Park | Xuzhou | Scenery | Exercise | Recreation | Attractive | Xuzhou City | Place |
| | Frequency | 28 | 11 | 11 | 11 | 10 | 9 | 8 | 8 | 7 | 7 |
| **Zhushan relics Park** | Characteristics words | Park | Environment | Waterfall | Scenery | Place | Xuzhou | Jinlong Lake | Landscape | Worth watching | Scenery |
| | Frequency | 23 | 11 | 10 | 10 | 8 | 7 | 7 | 6 | 6 | 6 |
| **Jiulong Lake Park** | Characteristics words | Park | Place | Jiulong Lake | Recreation | Environment | Walking | Somewhere | Recreation | Suitable | Scenery |
| | Frequency | 22 | 12 | 11 | 8 | 8 | 7 | 6 | 6 | 6 | 6 |
| **Kuaizai Ting Park** | Characteristics words | Park | Xuzhou | Kuaizai Ting | Recreation | Place | Small | Citizen | Free | Walking | Worth |
| | Frequency | 22 | 13 | 8 | 8 | 7 | 4 | 4 | 4 | 4 | 4 |
| **Dalong Lake Park** | Characteristics words | Dalong Lake | Grassland | Scenery | Suitable | Xuzhou | Recreation | Scenery | New city | Environment | kiteflying |
| | Frequency | 20 | 14 | 14 | 13 | 12 | 12 | 12 | 11 | 11 | 10 |
| **Jinlong Lake Park** | Characteristics words | Scenery | Xuzhou | Jinlong Lake | Place | Scenery | Environment | Walking | Suitable | Graceful | Recreation |
| | Frequency | 18 | 14 | 13 | 12 | 11 | 9 | 8 | 8 | 7 | 7 |
| **Kuishan Park** | Characteristics words | Park | Scenery | Place | Environment | Fresh Air | Graceful | Free | Suitable | Walking | Citizen |
| | Frequency | 17 | 8 | 7 | 7 | 7 | 6 | 6 | 5 | 4 | 3 |
| **Wuming Shan Park** | Characteristics words | Walking | Fresh Air | Place | Park | Clean | Free | Interesting | Scenery | Environment | Sightseeing |
| | Frequency | 5 | 4 | 4 | 3 | 3 | 3 | 3 | 3 | 2 | 2 |

The high-frequency words used to describe 19 parks were processed using the online Word cloud tool WordArt [31], and a word cloud map of all the parks was obtained (Figure 5). The results are as follows:

(1) The parks most frequently commented on included Yunlong Lake, Yunlong Mountain, Han Culture, and Guishan.

(2) The word cloud map reflects the impression that different parks leave on visitors, and these elements form the perceptual elements of each park. Of all visitors' perceived observations, the related keywords included: (a) "Nice", "spectacular", "beauty" and so on depict the aesthetic service of a park. (b) "Funny", "leisure", "walking", "entertainment", show the park's recreational service. (c) "Swimming", "running", "mountain climbing", "exercise" and so on reflect the park's sports services. (d) "Culture", "history", "relics" and so on represent the park's historical and cultural services.

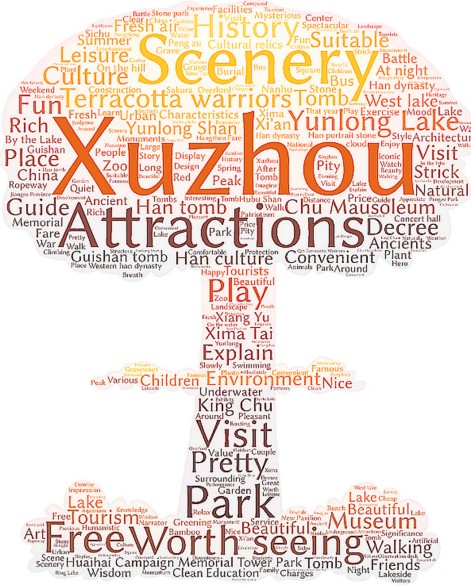

**Figure 5.** Word cloud map of Xuzhou parks. Note: The word cloud map was drawn based on the 230 most frequently used comment keywords left by visitors to Xuzhou parks. The more frequently used keywords are recorded in comments; more significant keywords are larger in the word cloud diagram.

### 4.2.2. Perception Discrepancy of Park's Cultural Services

Perceived differences were observed between the categories of cultural services in various urban parks. The comment records for parks with the characteristics of high quality, good location, and a long history received a greater number of comments documenting a perceived level of providing more cultural service types (Figure 6). For example, large parks, such as Yunlong Lake Park, Han Cultural Park, Yunlong Mountain Park, consistently had the most comments across all cultural service types, with respectively 2887, 2502, and 1171 documented keywords (including duplicate keywords). Meanwhile, the small size parks usually did not receive comments that documented the perception of cultural service types comprehensively with fewer comment records and a small number of related keywords. Examples include Kuaizai Ting (33 comment records), Kuishan (79 comment records), and Wuming Shan (48 comment records) parks, etc. These small parks mainly serve the local residents, so perceptions of cultural service types were relatively fixed.

Among the different cultural service types, aesthetic and recreation were the most readily perceived respectively. 13 and 12 park perception results relating to aesthetics and recreation services, keyword frequency for these services included more than 20% of the total records. Perceptions of inspiration and educational services were closely related to the theme of each park. A major category of cultural services in special parks was particularly obvious. For example, the educational service keywords were present in 67.81% of all records for Huaihai Campaign Memorial Tower Park. Meanwhile,

for Guishan, Hubu Shan, Han Cultural parks with historical sites, the number of cultural heritage keywords also reached 67.43%, 66.97%, and 63.67%, respectively, with similar frequencies for other parks at historical sites.

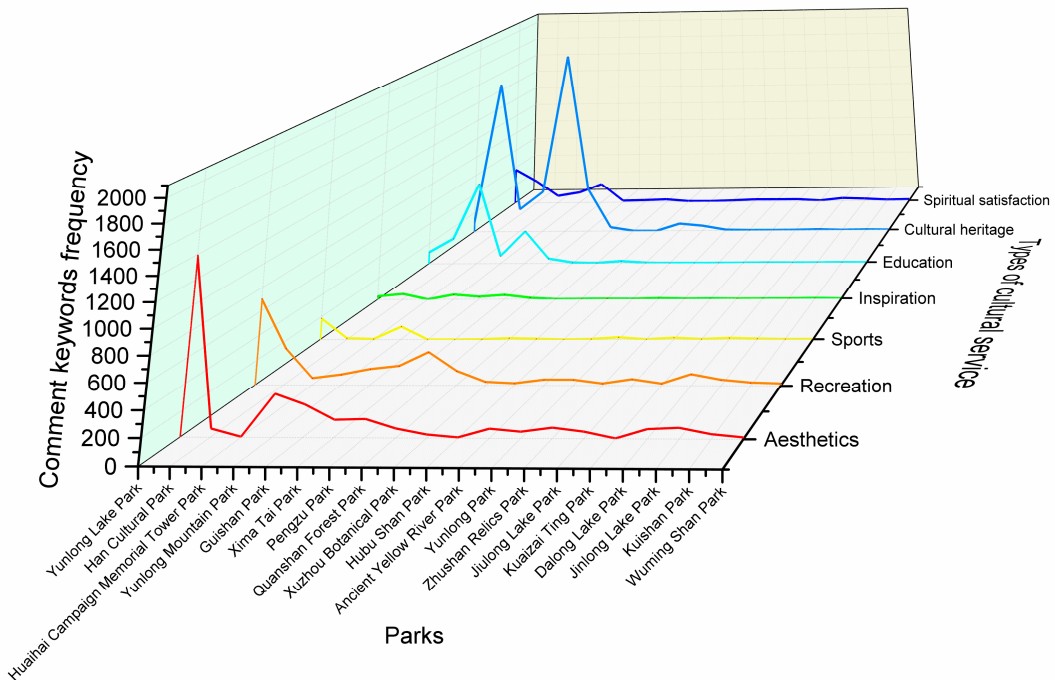

**Figure 6.** Comment keywords distribution for cultural services in each park.

### 4.2.3. Correlations between Different Cultural Service Categories

We determined the correlations between different service categories by calculating the Pearson correlation coefficient between each pair of cultural ecosystem services. We found that the aesthetic category was significantly correlated with most other service types, including sports (0.91), recreation (0.887), and spiritual satisfaction (0.804). Recreation and spiritual satisfaction were also significantly correlated (0.845), as were recreation and sports (0.744), sports and spiritual satisfaction (0.719), inspiration and cultural heritage (0.658), and spiritual satisfaction and inspiration (0.649). However, sports and cultural heritage were negatively correlated (−0.038) (Table 3).

**Table 3.** Pearson correlation coefficients between the various pairs of cultural ecosystem service categories.

| Categories | Aesthetic | Recreation | Sports | Inspiration | Education | Cultural Heritage | Spiritual Satisfaction |
|---|---|---|---|---|---|---|---|
| Aesthetic | 1 | **0.887 \*\*** | **0.910 \*\*** | 0.404 | 0.039 | 0.054 | **0.804 \*\*** |
| Recreation | **0.887 \*\*** | 1 | *0.744 \*\** | 0.539 * | 0.116 | 0.217 | **0.845 \*\*** |
| Sports | **0.910 \*\*** | *0.744 \*\** | 1 | 0.455 | 0.016 | −0.038 | *0.719 \*\** |
| Inspiration | 0.404 | 0.539 * | 0.455 | 1 | 0.172 | *0.658 \*\** | *0.649 \*\** |
| Education | 0.039 | 0.116 | 0.016 | 0.172 | 1 | 0.435 | 0.371 |
| Cultural heritage | 0.054 | 0.217 | −0.038 | *0.658 \*\** | 0.435 | 1 | 0.571 * |
| Spiritual satisfaction | **0.804 \*\*** | **0.845 \*\*** | *0.719 \*\** | *0.649 \*\** | 0.371 | 0.571 * | 1 |

(n = 19; * $p < 0.05$; ** $p < 0.01$); Significant correlations (>0.80) are bolded; strong correlations (0.6–0.8) are italicized.

### 4.3. Spatial Distribution of Cultural Ecosystem Services

The map reveals the perceived variation in the spatial distribution of the perception of the seven types of cultural ecosystem services based on park visitor comments, and also shows the representative cultural service provided by each park (Figure 7). As the figure shows, the hotspots for the perception of cultural services by park visitors in the Xuzhou area located in the western, central,

eastern, and northwestern parts of the city. Judging from the spatial distribution of the perception of cultural services:

(1) The parks that were perceived as aesthetic were clustered in the western part of the city, such as in Yunlong Lake and Yunlong Mountain parks.

(2) The hotspot area for the perception of recreation values was located in the western, central, and eastern parks of the city, while the representative parks were Kuaizai Ting, Quanshan Forest Park, Kuishan, and Dalong Lake Park.

(3) The hotspot area of sports service perception was also located in Yunlong Lake and Yunlong Mountain parks in the western part of the city.

(4) Inspiration services were concentrated in the western areas of Yunlong Lake and Yunlong Mountain, the central areas of Xima Tai, Pengzu, and the Eastern Han Cultural Park, as well as in the northwestern area of Guishan Park.

(5) Educational services were perceived in the Huai Hai Campaign Memorial Tower Park in the middle of the city, in Han Cultural Park in the east, in the Guishan Park and Xuzhou Botanical Park in the northwest.

(6) Cultural heritage services were perceived in the ancient imperial burial area known as Guishan Park in the northwestern part of the city, in Han Cultural Park in the east, in Huai Hai Campaign Memorial Tower Park in the city center, and in the Yunlong Mountain and Xima Tai park areas.

(7) The hotspots for the perception of the spiritual services by park visitors were in the parks of Yunlong Lake and Yunlong Mountain in the western part of the city, in the center of the city at Huai Hai Campaign Memorial Tower Park, in the Han Culture Park in the east, and in Guishan Park in the northwest. Overall, the perception of general urban cultural services formed hotspot areas in the western, middle, eastern, and northwest regions as a group, so that the surrounding adjacent region had synergistic characteristics.

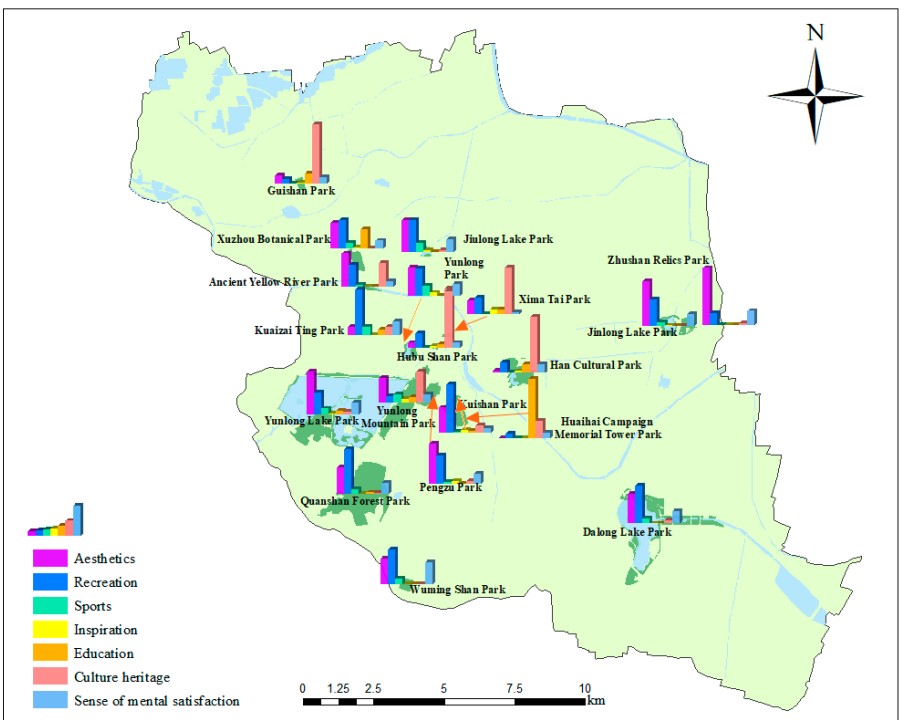

**Figure 7.** Spatial distributions of cultural ecosystem services of each park in the study area. Note: The bars show the proportions of each cultural ecosystem service category based on park comment records.

## 5. Discussion

### 5.1. Reliability of Perception Results

Among the published research results on the perception of cultural services by park visitors, recreational (85.8%) and aesthetic (55.9%) were obvious, but inspiration was very rarely commented on [32]. The results of a study related to the perception of specific values showed that spiritual, aesthetic, and recreational values occupied the top three positions among the villagers, sequentially [33,34]. Especially at larger-scale studies (e.g., regional, national, or international level), recreation is usually perceived as the main cultural ecosystem service [35]. For example, the perceptions of farmers in the Upper Parana Atlantic Forest Rainforest of Paraguay related to the value of rainforest were focused primarily on recreation and culture [36]. Creating a deeper understanding of past values through the archival analysis of photographs and oral histories would not only give researchers crucial background information but might also interest local people in mangrove conservation issues by increasing the sense of place, inspiration, and cultural heritage [37]. Site characteristics in urban parks directly influence visitors' perception. Sometimes, special facilities, such as benches are likely interpreted as indicators for recreational service [38].

The results of this study showed that cultural heritage, aesthetic, and recreation services were the most readily perceived in the cultural service perception for urban parks, reaching 36.25%, 21.31%, and 16.51% of the total number of perceptual keyword records. This is closely linked to the history of Xuzhou, a city with a history reaching back more than 2600 years, the emergence of a number of emperors from the area, the construction of a large number of mausoleums in the mountains, and some areas with terracotta warriors that are also preserved under the lake because of human activities. As a result, cultural heritage accounts for the highest proportion of the perception in the keywords used by visitors to Xuzhou. Aesthetics and recreation perception were similar to those reported in the existing literature and were the most easily perceived services. In primarily aesthetic parks, keywords related to recreation and spiritual satisfaction appeared more frequently; indeed, the correlations among these categories are stronger than between all other pairs of categories.

### 5.2. Value of the Research Results

The online comments of visitors employed here express their subjective perception and reveal their emotional feelings towards each park they have visited. For example, some parks received negative comments, which are valuable to managers and provide targeted information that can improve programs. Parks that had perceived aesthetic services were said to require better maintenance of trees, flowers, and other elements to ensure the sustained quality of aesthetic landscapes, while parks that provided recreational services were said to require maintenance of their recreational facilities to enhance visitors' satisfaction.

The main types of cultural ecosystem services provided by a park are closely related to the characteristics of that park. That is, parks with water features or forests were strongly correlated with aesthetic and recreation services, including Yunlong Lake Park, Quanshan Forest Park, Jiulong Lake Park, and Dalong Lake Park. Parks with primarily cultural heritage services have cultural relics more than one thousand years, including Guishan Park, Han Cultural Park, Hubu Shan Park, and Xima Tai Park. The park with education as the main service type is the Huai Hai Campaign Memorial Tower Park, which was built to commemorate the Huaihai Campaign, one of the three major battles in modern China. This park is also a famous red revolutionary education base in China, and thus had the highest proportion of educational keywords. The negative correlation between sports and cultural heritage services was due to the fact that cultural parks tend to charge admission, and only areas adjacent to the park are free of charge. Generally, visitors preferred to visit the free-of-charge mountain and lake parks for exercise and fitness activities.

The mapping of cultural service hotspot areas obtained using perceptual results reveals details about the agglomeration status of different services and the spatial distribution characteristics of

cultural service values. This information provides a reference for park managers related to park services, maintenance, management, and so on, while also providing a basis for integrating the ecosystem service concept into the planning and design of parks, as well as determining the main cultural services provided by each park.

*5.3. Methodological Limitations and Future Research*

Unlike the information obtained from questionnaires, data based on online comments from social networks have unique advantages, such as convenient data collection, a large amount of data available, and strong randomness, so this provides a new data source for perceptual research related to cultural ecosystem services. This article offered an opportunity to illustrate how the perception of cultural ecosystem services of urban parks could be analyzed through online comment data in China. Meanwhile, many such network sites exist abroad, such as TripAdvisor, Travelocity, Booking.com, Hotels.com and, so on, researchers also could analysis the network comment data by following this method from different perspectives. However, some limitations exist when using social network data, these can be mainly classified into three ways as follows:

(1) Information about individual commentators is difficult to collect. Commentator data, including gender, age, occupation, and income, are very meaningful for the study of cultural service perception. Analyzing this type of data is an important part of understanding the variations in perceived information. The lack of such data is not conducive to the subdivision of research data based on perceptions.

(2) The classification of keywords. Comment content can be split into keywords relatively easily, but it is difficult to judge the type and value of a park's cultural service, the environment of the cultural service in the park, and its effect on perception based on the keywords.

(3) Comprehensiveness of comment data issues. Online comments from social networks can only reflect the perceptions of visitors who comment on the Internet and, thus, the perceptions of several groups were excluded, including older people, children, and others that do not use the Internet regularly.

In future research, we need to explore methods that can be used to overcome the limitations of online comments, and further study the methods employed when using this kind of information in the evaluation of the cultural services provided by urban parks. At the same time, comments can also be used to relate the cultural ecosystem services to certain elements or the spatial extent of features within the parks. It would be very interesting to analyze areas inside the park areas to identify hotspots of certain cultural ecosystem services.

## 6. Conclusions

This paper presents a framework used for perception analysis of the cultural services provided by urban parks using social network data, which can be used to extract the online comment content of park visitors and analyze it in a perceptual way. Using semantic mining software, the text of sentences was split based on the formation of words and phrases, and the high-frequency keywords were extracted. Using the Delphi method, the keywords related to cultural service were classified. Then, cultural ecosystem service categories for the urban parks were devised, and the perceptual relationship between the keywords and the cultural service type was established. Taking the 19 parks in Xuzhou as an example, the empirical analysis showed that the perception of park visitors can reflect the types and differences of cultural services available in these parks. With the development of the network era, social network data are becoming more and more abundant. Through data mining technology, these data can be applied to the recognition of service categories in urban cultural ecosystems, and in evaluations of park worth. This may greatly reduce the cost of acquiring this type of data and improve the efficiency of evaluation methods.

**Author Contributions:** P.D. analyzed the data, wrote the original draft and prepared figures. S.Z. reviewed the manuscript. Z.C. and H.H. collected the data. Y.G. prepared the tables. All authors read and approved the final manuscript.

**Funding:** This research was funded by [Fundamental Research Funds for the Central Universities] grant number [2017XKZD14] and the APC was funded by [Fundamental Research Funds for the Central Universities (grant no. 2017XKZD14)].

**Conflicts of Interest:** The authors declare no competing interests.

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
