# Peer review of "Perceptions of Cultural Ecosystem Services in Urban Parks Based on Social Network Data"

_sustainability, doi:10.3390/su11195386_

Round 1
Reviewer 1 Report
Dear Authors,
thank you very much for this very interesting paper and I learned a lot. I like the paper very much and suggest to publish it after some revisions. I suggest to work and address the following points:
Some reflections on the rating of the experiences - you do not consider, make use or correlate them later in the paper.
Figure 1: Metric scale in km in the map would be good (city map left)
Lines 97-98: It would be good to learn a little bit more why you opted for these three portals. I guess they are the most popular ones. If you could describe this very briefly and also provide their user numbers if possible. If someone wants to do a similar study, this would support to reflect which portals might be the most useful ones to conduct a similar study.
Lines 134-135: I do not quite understand what you mean with this. Did you also conduct a literature survey or did you use Vaz et al. 2018 as your main source for classification? Needs clarification.
Delphi rounds: Very interesting approach - I learned a lot here.
Lines 152-156: So you did a three round Delphi Expert round or did you have more rounds? It would be good to mention how many Delphi rounds you did somewhere in the text.
Lines 168-169:
Some were called "cultural service", "information function", "delightful and satisfaction", "social and cultural achievement", or "spiritual
demand". This would need a citation.
Table 1: Last right bottom field: "Shock": Is this really the right word when translating it into English, since it has a more negative connotation? Perhaps "striking" might be the better term - please check.
Figure 4:
This word art needs some explanation in the text section before to understand it. You have the continents and the oceans. How do they relate? Do the continents reflect the different parks or word groups? Please explain it in the text then. If this is just random outcome of Wordle, then a different arrangement or a different design might be better not to confuse the readers.
Figure 5:
More a comment: Perhaps a two axis graph would be more suitable with the x axis (different parks) at the bottom and a y axis at the left side - you do not weight things here which is below a given average or so with this graph here that might be suggested by this type of graph - otherwise, it needs a bit of description in the previous text section.
Line 259 ff:
Just a point of reflection or comment of me - perhaps for further work: can the cultural services also be related to certain elements or spatial extent of features within the parks? This would be very interesting to analyze inside the park areas for hotspots of certain cultural services.
Line 290ff: Discussion section:
The first section is a bit short - I miss a bit more discussion and reflection here to compare your results to other work (it does not need to be extensive or perhaps, mention that there is more literature and you use only these selected papers since they might be comprehensive or exemplary for others).
Reviewer 2 Report
Good quality paper, presenting new method of assessing ecosystem services, based on social media data.
Overall remarks:
Abstract is quite clear and properly describe the idea, place and topic. Methods and results are mentioned in the abstract as well, properly describing the matter and goals of the paper.
Introduction is well presented and shows the merit of a paper. In the section of Methods and materials the subsection of study area and data sources, as well as conceptual model of research are properly designed and well shown. Maybe subsection “Analysis of online comment data” needs to be presented a bit clearer and better organised. First: the Figure 3 could be extend, including and describing all the 10 steps of research design. Or please prepare additional scheme, presenting all the method. Second: the first paragraph of “results” seems to be a mixture between methods and results, please concentrate here of the results only, adding appropriate piece of information into the “methods”.
Results are highly interesting, and well described. I like word cloud and tables. I do not know I understand properly all the correlations. They are very nice but sometimes obvious – e.g. sport is not usually localised near the heritage, so people do not perceive it together.
Discussion is comprehensive, Authors referred their results very broadly, dividing discussion into three parts, and comparing them to interesting sort of cited literature in first part.
Seemingly all the interesting conclusions are “sold” in the discussion, because conclusions are very laconic and repeats part of introductions. In my opinion it will be smart to share some statements from the discussion between discussion and conclusions, especially “Methodological limitations and future design”.
Detailed remarks
Line 132-133 How many experts?
Line 243 The graph (figure 5) could be bigger.
